# Functionalization of Nanoparticulate Drug Delivery Systems and Its Influence in Cancer Therapy

**DOI:** 10.3390/pharmaceutics14051113

**Published:** 2022-05-23

**Authors:** Theodora Amanda Seidu, Perpetua Takunda Kutoka, Dorothy Owusu Asante, Muhammad Asim Farooq, Raphael N. Alolga, Wang Bo

**Affiliations:** 1Department of Pharmaceutics, School of Pharmacy, China Pharmaceutical University, Nanjing 211198, China; stheodoraamanda@gmail.com (T.A.S.); 3719019002@stu.cpu.edu.cn (P.T.K.); 2School of Basic Medicine and Clinical Pharmacy, China Pharmaceutical University, Nanjing 211198, China; senamdor@gmail.com; 3Drug Delivery, Disposition and Dynamics, Monash Institute of Pharmaceutical Sciences, Monash University, Parkville, VIC 3052, Australia; muhammad.farooq1@monash.edu; 4State Key Laboratory of Natural Medicines, Department of Pharmacognosy, China Pharmaceutical University, No. 639 Longmian Road, Nanjing 211198, China

**Keywords:** surface functionalization, multifunctional nanoparticles (MNPs), cancer therapy, liposomes, dendrimers, mesoporous silica NPs

## Abstract

Research into the application of nanocarriers in the delivery of cancer-fighting drugs has been a promising research area for decades. On the other hand, their cytotoxic effects on cells, low uptake efficiency, and therapeutic resistance have limited their therapeutic use. However, the urgency of pressing healthcare needs has resulted in the functionalization of nanoparticles’ (NPs) physicochemical properties to improve clinical outcomes of new, old, and repurposed drugs. This article reviews recent research on methods for targeting functionalized nanoparticles to the tumor microenvironment (TME). Additionally, the use of relevant engineering techniques for surface functionalization of nanocarriers (liposomes, dendrimers, and mesoporous silica) and their critical roles in overcoming the current limitations in cancer therapy—targeting ligands used for targeted delivery, stimuli strategies, and multifunctional nanoparticles—were all reviewed. The limitations and future perspectives of functionalized nanoparticles were also finally discussed. Using relevant keywords, published scientific literature from all credible sources was retrieved. A quick search of the literature yielded almost 400 publications. The subject matter of this review was addressed adequately using an inclusion/exclusion criterion. The content of this review provides a reasonable basis for further studies to fully exploit the potential of these nanoparticles in cancer therapy.

## 1. Introduction

Cancer could be a comprehensive term that includes a broad run of ailments that can adversely impact any part of the body. Breast, colon, lung, rectum, prostate, skin (nonmelanoma), and stomach cancers are the most common new cases around the world, accounting for nearly 10 million passings in 2020 [1]. It is expected that within the following two decades, new worldwide cancer cases will increase to 22 million [2]. One of the primary determinants is the inability of antitumor drugs to be delivered selectively to cancerous tissue. High systemic antineoplastic agent exposure often results in dose-limiting toxicity. As a result, a targeted delivery system is critical for overcoming current limitations in cancer therapy. The intricate biology of tumor cells with several biological barriers, such as mononuclear scavenger cell uptake; extravasation via the vascular endothelial membrane; and physiological features such as hypoxemia, low pH, and elevated pressure of interstitial fluid, emphasizes the necessity to invent and formulate an effective antitumor drug delivery system [3].

Therefore, an idealized drug delivery system (DDS) must possess two components, spatial placement (the potential to target) and temporal delivery (controllability of the release) of the drug [4]. The ability of a DDS to target and have a controlled release of cargo will enable an increase in the drug’s effectiveness and minimize adverse effects [5,6]. The most widely used drug delivery platform is nanoparticles (NPs), classified into two types—organic and inorganic NPs—with some popular ones being lipid-based NPs, polymeric NPs, protein NPs, and inorganic NPs. Nanoparticulate drug delivery systems (NPDDSs) demonstrate great potential as DDSs. Due to their nano size, increased surface-to-volume ratio, and advantageous physicochemical properties, they can modulate the pharmacodynamics and pharmacokinetic profiles of drugs by improving their therapeutic index [7]. They also express different but preferable chemical, physical, and biological effects [8]. NPDDSs are continually investigated to solve the deviance in conventional drug delivery systems. Many drugs possess a hydrophobic element; therefore, they have precipitation issues in higher concentrations and, although excipients are designed to prevent drug precipitation, they come with toxicity as a setback [9]. Due to the nature of NPDDS, they provide an amphipathic environment for drugs, thereby enhancing drug solubility, solving these issues of precipitation, and increasing their bioavailability [10]. Liposomes can simultaneously convey both hydrophilic and hydrophobic drugs due to the hydrophilic internal structure of liposomes, which significantly advances drug loading effectiveness [11]. Additionally, liposomes’ cell membrane architecture facilitates effective cell affinity and greatly boosts cellular uptake.

In recent years, inorganic NPs have also been extensively used for theragnostic cancer treatment due to their outstanding chemical and physical features. A typical inorganic nanoparticle is mesoporous silica NP, which has high drug loading instigated by surface pores filled with cargo. However, the body’s inability to metabolize the inorganic materials can lead to severe tissue injury. Functionalization by adding organic moieties to curb this setback (See in Figure 1) is, therefore, necessary [12]. Recently, the focal point in drug delivery technology has been the designs/functionalization and applications of NPDDS; for example, polymer and surface conjugations are being explored [13,14].

NPDDS are modified using various techniques to serve as diagnostics or therapeutic vehicles for many diseases [15]. Functionalization of NPs includes the modification of NPs with various targeting ligands, diagnostic agents, imaging agents, biomolecules, oligonucleotides, peptides, antibodies, stimuli-sensitive ligands, etc. to improve their properties or introduce new features to enable targeting with accuracy [16]. Most functionalized nanoparticles possess improved physicochemical properties, enhanced targeting ability, bioavailability, biocompatibility, anticorrosiveness, antiagglomerative, and noninvasive characteristics. An increasing number of researches are being undertaken to functionalize nanoparticles to enhance their gross efficiency and modality [14,15,17,18].

The following objectives guide the writing of this review article: to explore the recent strategies for targeting nanocarriers at malignant tumor sites and their benefits, relevant surface functionalization approaches in the design and synthesis of agile nanocarriers, and how functionalized nanocarriers improve anticancer drug delivery to the required location. Finally, the shortcomings of functionalized nanocarriers are mentioned and their prospects.

## 2. Nanoparticles and Their Classifications

The two main classifications of nanoparticles are organic and inorganic nanoparticles. Liposomes, dendrimers, and polymeric nanoparticles are examples of organic nanoparticles. Inorganic nanoparticles include mesoporous silica, gold NPs, carbon nanotubes, etc. [19,20]. Figure 2 presents the differences and applications of the two main classes of nanoparticles. The methods used to obtain nanoparticles significantly impact their morphology; size; structure; electrochemical, physicochemical, optical, and electrical properties; and general performance in cancer therapy [19].

Nanotechnology is a promising and rapidly developing field in the pharmaceutical and medicinal industries. As drug delivery systems, nanoparticles provide numerous benefits, including improved efficacy and lowered adverse side effects [21]. One of the most well-known areas of nanotechnology research is nanomedicine. It makes use of nanotechnology to create highly targeted pharmaceutical interventions for diagnosis, treatment, and possible prevention of diseases [22]. For the last few decades, there has been a surge in nanomedicine research, which is now being translated into operational commercialization all over the globe, ultimately resulting in the marketing of a wide range of products. DDSs now dominate nanomedicine and are estimated to account for more than 75 percent of cumulative sales [23]. The diameters of nanoparticles range between 10 and 1000 nm. The API is either encapsulated, entrapped, dissolved, or linked to the nanoparticle matrix [24]. By modifying the fabrication technique, nanoparticles can be developed. Nanoparticles have been demonstrated to be efficient delivery vehicles. There are numerous applications for nanoparticulate DDSs, including gene, cancer, AIDS, and radiation therapies. It can also transport proteins, antibiotics, and vaccines, as well as act as vesicles that are capable of crossing the blood–brain barrier [25]. The primary goals of nanoparticle architecture as a delivery platform are to regulate particle size, surface characteristics, drug delivery, and API release to establish site-specific drug activity with an optimal treatment rate and dosage regimen.

## 3. Liposomal Nanoparticulate DDS

Conventional liposomes usually have the drawback of being subsumed by the reticuloendothelial system (RES) and lead to short-term circulation [11]. This process (RES) occurs when opsonin (Serum protein) perceives them as foreign substances, and they are disrupted by immune cells. To overcome this setback (short circulation time), the liposome surface is plated with PEG to extend circulation by magnifying repelling forces between liposomes and serum elements; this forms the class of liposomes called PEGylated liposomes [26]. Liposomes can be grafted diversely with targeting ligands (ligand-targeted liposomes); examples include peptides and antibody fragments to implement diverse surface engineering strategies for targeting at requisite tumor sites using overexpressed receptors [27]. Immunoliposomes were also produced by chemically attaching antibodies or fragments of antibodies to the liposomal surface, leading to solid precision for their target antigens. In cancer treatment, the use of targeted liposomes formulated using a range of different engineering techniques has proven minimal off-target effects on healthy tissues [28,29,30].

Multifunctional liposomes are considered the advanced form of the single-functionality liposomes and have the potential to overcome the setbacks faced by the single-functionalized liposomes. A variety of functionalities are incorporated using various functionalization techniques and modifications. Liposomes having two ligands (dual-functional) [31], liposomes having two ligands with two anticancer drugs [32], liposomes having a targeting ligand with an imaging agent [33], theragnostic liposomes containing an imaging agent and therapeutic agent [34], etc. have been reported. Liposomes are used in cancer therapy [35] and other specialties such as mycosis (fungal infection) [36], gene transfer therapy [37], and vaccine nanocarriers [38,39].

### 3.1. Functionalization of Liposomes Using Targeted Ligands

Liposomes are imbued with varied targeting ligands using one or a combination of surface engineering strategies [40]. Ligands are most often covalently linked to liposomes via interactions between reactive groups on the liposome’s surface and particular groups within the ligand. Targeting ligands could be covalently or noncovalently incorporated to customize a targeted liposomal system for cancer treatment. Liposomes are functionalized with diverse targeting ligands via three reactions: amide bond formation involving carboxyl and amino groups, disulfide bond creation by reacting pyridyldithiols and thiol group, and thioether bond formed by reacting maleimide (MAL) and thiol group [40,41]. Preformed liposomes with reactive groups on their surfaces, such as MAL, were ligated with PEG derivatives in the postcoating technique [42]. PEG–lipid micelles were used in one technique to incorporate PEG–lipid conjugates into the liposome membrane without disrupting the liposomes. The conjugation reaction between thiol on the surface of ligands and MAL on liposomes is a critical surface engineering approach for liposomes with a suitable targeting ligand [43].

Targeting ligands commonly used are antibodies or antibody fragments, cell-targeting peptides or cell-penetrating peptides, and small molecules such as folate, based on their use in tumor therapy and aptamers. Some are shown in Table 1.

### 3.2. Functionalization of Liposomes Using Antibody

Surface alteration of liposomes with antibodies can be achieved by linking antibodies or their fragments to the surfaces of liposomes to produce immunoliposomes using several methods [58]. One method uses a covalent link between an antibody or fragments of antibody with liposomal lipids. Another method is to chemically modify the liposome to enhance its hydrophobicity by adding an appropriate substituent, typically resulting in a greater affinity for the liposome’s bilayers [59]. A recent strategy is to use the antibody fragment (e.g., fragment antibody-binding (Fab) or single-chain fragment variable (scFv)) in place of the entire antibody to avoid the risk of inactivation of the antibody or initiation of the immune response during surface functionalization and to reduce particle sizes to enhance efficient delivery of anticancer agents. There are several reviews on the techniques for antibody derivatization and the development of reactive groups to be coupled with lipids or preformed stealth liposome [60]. Numerous reagents are also employed in thiolating antibodies (e.g., 2-Iminothiolane (Traut’s reagent) to produce sulfhydryl group). The antibody moieties contain groups modified for active targeting, such as thiol, carboxyl, and amine groups. Usage of a sulfhydryl group in connecting a thiolated antibody with a lipid-containing reactive group, i.e., MAL, is the primary strategy and has been widely studied in the literature. The sulfhydryl group is susceptible to oxidation and can be prevented by substituting oxygen with ethylenediaminetetraacetic acid [60]. Liposome surface functionalization with fragments of antibody has been accomplished using the same methods. Carbonic anhydrase, CA, are zinc metalloenzymes found on the surface of red blood cells [41]. Many tumors exhibit hypoxia. In cancer microenvironments, hypoxia is associated with reduced extracellular pH (around pH 6.5). Due to the lack of oxygen, carbonic anhydrases (Cas IX and Cas XII) are overexpressed in several solid tumors, predominantly lung and brain tumors. CA IX is more active than CA XII [61]. In a study, CA IX targeted immunoliposomes encapsulated with docetaxel was fabricated. The in vitro binding and cell uptake of liposomes to A549 cells (CA-IX positive and CA-IX negative) was explored by means of fluorescence-based flow cytometry. Results showed increased uptake in CA-IX positive A549 cells [62].

### 3.3. Functionalization of Liposomes with Peptides

Liposomes’ surfaces are also functionalized with peptides, primarily through covalent and noncovalent bonding [63]. Peptides are covalently linked to liposomes via various linkages, including the MAL, peptide, sulfanyl, disulfide, and phosphatidylethanolamine-linker. Presently, disulfide and thioester links have been extensively studied. Noncovalent linkage has been used to attach amphipathic peptides to liposomes. Peptides used for the surface modification of nanocarriers can be classified into two types: cell-penetrating peptides (CPP) and cell-targeting peptides (CTP), which are nonspecific and receptor-specific (directly bind and increase internalization) respectively [63]. Paclitaxel-encapsulated T7 targeted liposomes inhibited tumor growth in ovarian-cancer-bearing mice more effectively than nonfunctionalized liposomes and free drug [64]. Human glioma cell growth was inhibited by doxorubicin-containing cyclic RGD peptide-modified liposomes (U87MG cell-line). The cyclic RGD binds to integrins, which are overexpressed in various tumors. Integrins are also highly specific for cyclic RGD [65]. PEGylated liposomes with CPP and an acid-sensitive hydrazone bond was made. It was discovered that a 4 percent CPP-to-lipid ratio resulted in greater liposome internalization efficiency into targeted compartments [66]. Patra et al. developed liposomes comprising carbon dots and CPP (polyarginine) for the delivery of curcumin throughout the skin. The inclusion of carbon dots aided in the imaging of skin fluorescence [67].

### 3.4. Functionalization of Liposomes with Aptamers

Aptamers are oligonucleotide-based single-stranded DNA or RNA sequences that can target receptor sites on the facet of tumor tissues. Systematic ligand development using exponential-enrichment technology resulted in the production of aptamers with greater attraction for the molecules targeted [68]. Liposomes with a targeting aptamer ligand attached to their surface and containing antitumor cisplatin have been reported. Nucleolins (NLC) were the aptamer’s preferred target. Aptamer-controlled liposomes were formed by incorporating the cholesterol-labeled aptamer with the other liposomal components prior to hydration. Parallel to free drug and nontargeted liposomes (loaded with drug), aptamer-targeted liposomes were found to have increased antiproliferative activity in breast cancer cells (MCF-7) overexpressing NCL. This research shows that breast cancer cells that overexpress NCL were deliberately targeted. An aptamer-based liposomal preparation was proposed to combat multidrug resistance (MDR) in breast cancer, and P-gp transporter is overexpressed in MDR metastatic breast cancer cells [69]. This overexpression can be reduced by using siRNA to silence it. Liposomes loaded with siRNA and loaded with aptamer A6 (which has attraction for HER2 receptors on breast cancer cells) were prepared to improve siRNA delivery into breast cancer cells. Aptamer A6 was brooded with prefabricated liposomes that contained a PEG–MAL, MAL group that could be conjugated with the aptamer. A6-directed liposomes were produced during the incubation period. According to the study’s findings, aptamer-directed liposomes could convey siRNA (targeted at P-gp) into cancerous breast cells to counter chemoresistance [70].

### 3.5. Liposome Functionalization with Small Molecules

In cancer therapeutics, small molecules such as folate, affibody, carbohydrate, and others have been used as targeting moieties for surface functionalization of liposomes. Aimed at cervical cancer therapy, folate-targeted liposomes encapsulating imatinib have been formulated. The film hydration approach was used to create liposomes, including a folate–lipid conjugate during the lipid film forming step. Transmembrane pH gradient was used to load imatinib into liposomes. Folate-targeted liposomes decreased the IC50 value of cervical tumors (HeLa-cells) sixfold, from 910 M to 150 M [71]. It has been reported that HER2-targeting affibody (Z00477)2-Cys coupled liposomes are used to treat breast cancer, i.e., TUBO cloned cells and SK-BR-3 cells. Thioether linkage was used to reduce affibody and conjugate it with DSPE-PEG-MAL micelles. Cisplatin was solvated to an aqueous phase during the formulation before being loaded into liposomes. The ethanol injection technique was used to create liposomes. Surface modification of cisplatin-containing liposomes with affibody-conjugated micelles was achieved after a 4-h incubation at 47 °C. Affisomes demonstrated increased cellular uptake and therapeutic effectiveness [72].

### 3.6. Dual-Ligand Functionalization of Liposomes

Surface-functionalized liposomes with two ligands has also been reported with promising outcomes. Kang et al. used chimeric-ligand aimed multifunctional liposomes, such as folate linked-peptide-1 and two ligand-directed liposomes, such as folate and peptide-1-targeted liposomes, using the film hydration technique. The foremost step is to create PEGylated liposomes with a terminal MAL group. The liposomes were then cultured with chimeric-ligands such as folate and peptide-1 ligand to produce chimeric-ligand driven liposomes and a double ligand guided liposome, respectively. The FITC-dextran-loaded multifunctional liposomes were tested on uterine-cervical cell lines (HeLa) and human-keratinocyte cell lines (HaCaT). Double ligand driven liposomes outperformed chimeric-ligand directed liposomes in cell uptake and cytotoxicity in the HeLa cell lines [73]. Zhang et al. also confirmed the development of targeted liposomes having two peptides (TfR and VEGFR2 specific peptides) and two antitumor agents (doxorubicin and vincristine), which enhanced drug delivery and optimized therapeutic efficacy in the brain (see results in Figure 3) [32]. Zong et al. also developed doxorubicin liposomes with two peptides (TAT and T7), which demonstrated improved therapeutic efficiency in glioma therapy in animals contrasted to one ligand dox liposomes and just dox [74].

### 3.7. Stimuli-Sensitive Liposomes

Stimuli cause liposomes to become unstable, resulting in the release of trapped payload. The primary stimuli used to enhance the delivery of anticancer drugs to the required site via liposomes include temperature, pH, magnetic field, ultrasound, light, redox, and enzymes [75].

#### 3.7.1. Temperature-Responsive Liposomes

Liposomes are constituted of thermosensitive lipids that are stable at 37 °C. Thermoresponsive liposomes were utilized to target cancer cells with an elevated temperature compared with the rest of the body. When the temperature was raised from 37 to 41 °C, the liposomes made of cholesterol and dipalmitoylphosphatidylcholine released approximately 80 percent of the encased methotrexate within 30 min. Temperature-sensitive liposomes are utilized in the commercial antitumor liposomal formula Thermodox^®^ (Celsion, Lawrenceville, NJ, USA). It constitutes 1-myristoyl-2-stearoyl-sn-glycerol-3-phosphocholine, which has a 40 °C transition temperature [76,77].

#### 3.7.2. pH-Sensitive Liposomes

pH-sensitive liposomes are stable at pH 7.5; however, changes in pH, such as those found in cancer tissues—that is, low pH—cause the encapsulated cargo to be released due to bilayer instability [78,79]. At the tumor site, the pH can drop to 5.7 [80]. 18:1 or DODAP DAP is a pH-sensitive lipid that is frequently employed to create pH-responsive liposomes [41]. In one study, antigenic peptides originating from ovalbumin and pH-responsive fusogenic polymer were loaded into liposomes to develop peptide vaccine-based cancer therapy, which resulted in decreased tumor volume [81].

#### 3.7.3. Magnetic-Field-Sensitive Liposomes

Magnetic-field-sensitive liposomes have iron oxide cores (magnetite and Fe_3_O_4_) that magnetize when exposed to an extrinsic magnetic field [79]. Magneto-liposomes containing 5-fluorouracil were created. The film hydration technique was exploited to produce liposomes. Evaporation under vacuum was utilized to create a lipid film of PC solution in chloroform, which was then hydrated with Fe_3_O_4_ suspension in water. Due to the hyperthermia effect, a magnetic field triggered the drug to be released in human colon cancer cells and tumor growth suppression was noticed [82]. In a study, liposomes comprising iron-oxide and methotrexate accumulated extra into target tissue in a model mouse when an extrinsic magnetic field was applied versus the same liposomes when no extrinsic magnetic field was applied [83].

#### 3.7.4. Ultrasound-Sensitive Liposomes

When tiny gas bubbles in liposomes are exposed to ultrasound waves, they generate echo sound, which allows for ultrasound imaging. Ultrasound waves can also break up liposome systems, allowing the drug to be released at the desired location [84]. Breast tumor progression was significantly inhibited in MDA-MB-231 tumor-bearing mice when doxorubicin liposomes with a CO_2_ bubble creating thermosensitive system were used instead of just thermosensitive doxorubicin liposomes without gas. Thermoresponsive liposomes proficient in damaging lipid bilayers are formed by rehydrating dried lipid films with citrate buffer and generating CO_2_ bubbles (300 mM, pH 4). Due to a synergy between burst drug release and hyperthermia-induced CO_2_ production, doxorubicin’s antitumor activity was increased. An ultrasound imaging technique was used to track the CO_2_ production caused by hyperthermia. In this report, the drug is discharged from liposomes as a side effect of hyperthermia, which induces CO_2_ production in the liposomes [85].

#### 3.7.5. Multiple Stimuli-Sensitive Liposomes

A recent phenomenon is the evolution of liposomal formulas that respond to multiple stimuli simultaneously. Many researchers are interested in customized nanocarriers cotriggered by multiple stimuli in various organisms (e.g., extracorporeal, tumor tissue, cell, subcellular organelles) as they can overcome sequential physiologic and pathologic change barriers to deliver diverse therapeutic “payloads” to the desired targets. Furthermore, DDSs that are sensitive to several stimuli provide an excellent platform for agent codelivery and reversing multidrug resistance [86]. Liposomes comprising a pH-responsive monomer (2-propyl acrylic acid) and a temperature-sensitive monomer (N-Isopropylacryamide (NIPAAm)) were developed utilizing p(NIPAAm-co-PAA) copolymer. Variations in pH and temperature impacted the release of doxorubicin. The film hydration approach was used to make liposomes, and doxorubicin was encapsulated using a pH gradient. MR-directed ultrasound was utilized on heat-defined tissues and activated local release of drug. The administration of these liposomes resulted in increased cytotoxicity in breast tumors. The goal of developing duple-sensitive liposomes was to reduce the side effects on healthy tissues [87]. A double stimuli-sensitive liposomal structure was established to tackle the issue of CPP deterioration in CPP-siRNA conjugates. A suitable strategy and a current surge for increasing the potential of cancer treatment is the production of liposomes that react to stimuli and have surfaces designed to aim at receptors overexpressed on the surface of tumor cells and in the tumor microenvironment.

Antibody-targeted thermoresponsive liposomes were developed, according to a previous article. As per the research, after surface modification with hCTMO1-antibody, the thermal and physicochemical characteristics of conventional thermoresponsive liposomes (TTSL) were retained. TTSL was made using the film hydration technique. By conjugating thiolated antibodies with the MAL group, DSPE-PEG-MAL-hCTMO1 micelles were produced. By incubating DSPE-PEG-MAL-hCTMO1 micelles with preformulated TTSL for one hour at 60 °C, the thermoresponsive, targeted liposomal structure was developed. Compared with conventional thermosensitive liposomes, the implementation of produced liposomal structure with heating at 42 °C for one hour led to elevated cellular uptake and cytotoxicity in breast tumors (MDA-MB-435 cells) overexpressing the MUC1 gene [88]. It is worth noting that developing stimuli-responsive liposomes with surfaces designed to target receptors primarily expressed on the surface of tumor cells or in the tumor microenvironment is a promising option and provides a new surge for maximizing the potential of cancer therapy [89].

## 4. Dendrimer Nanoparticulate DDS

Despite the widespread use of dendrimers in pharmaceutics and biomedicine, its complex basic features, such as toxicity due to the existence of a terminal NH2 group on the surface of dendrimer and fast clearing of dendrimer from the blood stream amid cancer treatment, restrain their function in the treatment of cancer [90,91,92]. Various studies have shown that the toxicity of dendrimers is influenced by their generation (size), development time, concentration, and the sort of terminal group on their exterior [93,94,95,96]. Chen et al. investigated the role of surface groups in dendrimer toxicity by functionalizing cationic melamine dendrimers with various attached groups such as amine, guanidine, sulfonate, phosphonate, and carboxylate. Cationic dendrimers had greater toxicity than anionic, polyethylene glycolylated (PEGylated) and neutral dendrimers [97]. Another group of scientists examined the cytotoxicity of just G5.0 PPI dendrimer versus dendrimer functionalized using carbohydrate and amino acid on cos-7 and HePG2 cell lines. They discovered a concentration- and time-related decline in cell viability and established that it could be attributed to positive charges on the dendrimer [98]. Surface functionalization appears to be the most effective approach for making dendrimer a suitable carrier in cancer treatment. Functionalization does not only minimize flaws in dendrimer structural design but also enhances dendrimer features and introduces novel ones. For instance, attaching neutral molecules on dendrimer surfaces covers/layers the positive charges, resulting in decreased cytotoxicity and increased vector circulation duration in biological systems [94,99,100].

Similarly, functionalization of dendrimers with different targeting molecules such as vitamins, antibodies, and PEG chains (See in Table 2) improves biocompatibility, increases transfection efficacy, elicits specific site delivery, and shows controlled and sustained release behavior in the preferred location of action [100]. Due to the multivalency trait of dendrimers, two or more targeted moieties can be conjugated on their surface simultaneously, and the production of multifunctional or bifunctional systems may be a better approach for increasing therapeutic efficacy and reducing nonspecific vector placement [101]. The biocompatibility and circulation period of dendrimers can also be increased by conjugating them with biocompatible moieties using three methods: (1) dendrimer functionalized with a biocompatible core—this method improves biodistribution and circulation time of dendrimers while increasing the possibility of surface functionalization; (2) dendrimer functionalized with biocompatible unit as a duplicating branch of the dendrimer—this type of dendrimer is nontoxic and water-soluble, making it a suitable carrier for drug delivery in cancer therapy; (3) surface-functionalized dendrimers with biocompatible moiety—this functionality may be the unsurpassed way to eliminate toxicity and escape from fast clearance [99,102].

### 4.1. Declining Toxicity of Dendrimers by PEGylation

Dendrimers are functionalized with PEG as a required procedure to reduce toxicity towards endothelium. Functionalization with the PEG chain caps the dendrimer’s positive charges and inhibits electrostatic interference amid the cell’s outer membrane and the dendrimer’s cationic surface. PEG chain could be directly attached or conjugated onto the facet of the dendrimer via a linker—usually to lessen steric hindrance—or attached to the surface of the dendrimer as a linker for other molecules (i.e., drug molecules) to be attached [103]. The protecting effects of PEGylation on dendrimer toxicity were studied by analyzing the cytotoxic effects of positively charged PAMAM dendrimers coated with 200 KD PEG chains. Owing to the capping of positive charges on their surface, PEGylated PAMAM dendrimers were less toxic than unmodified dendrimers. It was presumed that PEGylating could be an effective tactic for lowering toxicity and enhancing circulation by preventing reticuloendothelial system recognition [104]. PEGylation enhances biodistribution and pharmacokinetics, as well as drug-loading capacity. However, PEG chain conjugation on the dendrimer surface is affected by PEG length and total sum of arms, which impacts drug encapsulation efficiency of the dendrimer. Kojima et al. loaded antitumor agents inside PEGylated dendrimers and demonstrated that lengthier PEG arms had a higher drug-loading capability and increased stability of trapped drugs [105].

### 4.2. Declining Toxicity of Dendrimers by Acetylation

Another method for reducing dendrimer toxicity is the acetylation of dendrimers by conjugating an acetyl group onto the dendrimer, which causes the nullification of positive charges on the dendrimer surface [106]. Acetylation has many advantages over PEGylation, including high acetylation efficiency and ease, low amount of acetylation required, and minimized steric hindrance compared to the PEG chain [99,102]. Wang et al. synthesized acetylated PPI dendrimers with varying measures of an acetyl group and then loaded doxorubicin, sodium deoxycholate, and methotrexate sodium into the acetylated dendrimer. In A549 and MCF-7 cell lines, acetylated PPI dendrimer significantly reduced cytotoxicity. Increased acetylation ratio greater than 80 percent resulted in augmented drug loading, decreased toxicity, and enhanced pharmacokinetics. Doxorubicin and Methotrexate sodium cytotoxicity was remarkably reduced when coupled with increased acetylation of a PPI dendrimer [107]. Aside from PEGylation and acetylation approaches used to reduce toxicity of dendrimers, there are molecules such as carbohydrates, peptides, and amino acids (e.g., glycine and phenylalanine) that can significantly reduce toxicity by neutralizing positive charges on the dendrimer. Further, conjugating DNA and drugs onto dendrimer and formation of complexes owing to the covering of positive charges with negative charges of DNA reduces dendrimer toxicity [108].

Targeted delivery is one of the goals in cancer therapy; however, owing to unspecific binding in vivo, antitumor agents are not accumulated at the targeted sites [109]. Nevertheless, likened to normal cells, tumor cells overexpress specific receptors on their surfaces [110]; due to this, a strategy has been deduced by conjugating the nanocarriers with ligands that have an affinity for these overexpressed receptors, thereby accumulating antitumor agents at the tumor sites [90]. This approach results in tumor-specific targeting; it also hinders the reticuloendothelial system and increases permeability and retention (EPR) [111]. Ligands used for targeting in dendrimers are vitamins, peptides, antibodies, and aptamers [112]. The table below summarizes the various targeted ligands used to engineer dendrimer-targeted delivery systems in vivo and in vitro.


pharmaceutics-14-01113-t002_Table 2Table 2Targeting ligand-functionalized dendrimers summary.Nanodelivery SystemLigandsReceptorsDrug/DiseaseRef.PEG PAMAM dendrimer AS1411 (aptamer)NucleolinColon cancer (c26), HT29, CHO cells[111]PAMAM dendrimer–PEGFLT1(antibody)Vascular endothelial growth factorsreceptorGemcitabine, pancreatic cancer[113]PPI dendrimerFolic acid (vitamin)Folic acid receptorDoxorubicin, breast cancer (MCF-7 cell line)[90]PAMAM dendrimer (G4, G3.5)Biotin (vitamin)multivitaminTransporter(Na-dependent)Cisplatin, ovarian cancer (OVCAR-3, A2780, SKOV-3)[114]PAMAM dendrimerHyaluronic acid (glycosaminoglycan)CD44 receptor3,4-Diflluorobenzylidene curcumin, pancreatic cancer (MiaPaCa-2)[115]PAMAM dendrimer G5N-Acetyl galactose amine (NAcGal)(carbohydrate)Asialoglycoprotein receptor (ASGPR)liver cancer—HePG2[116]


Stimulus-responsive nanocarriers reduce the toxic effect of chemotherapeutic agents in cancer therapy by sensing environmental changes in biological conditions. Stimuli-responsive DDS improves drug release behavior and anticancer effects based on the specific stimuli and precise release site by allowing for a regulated release and easy application via a switch on–off technique [117]. Stimuli-responsive transporters are sensitive to various endogenic and exogenic stimuli, including decreased pH, increased glutathione concentrations, overexpression of specific enzymes, temperature, and light (See examples in Table 3). As a result, stimuli-responsive carrier systems enhance therapeutic effectiveness while lowering side effects.

### 4.3. Increasing Transfection of Dendrimers

High-generation dendrimers have a high transfection efficiency but have a high level of toxicity, whereas low-generation dendrimers have a low transfection efficiency but a low level of toxicity [123,124]. To reduce cytotoxicity and elude this demerit, molecules such as PEG and acetyl groups are conjugated to the dendrimer to neutralize the positive charges and reduce toxicity associated with the dendrimer’s cationic surface [99]. Positive charge neutralization on the dendrimer surface reduces the dendrimer’s buffering ability. Buffering capacity is determined by a primary amine group on the exterior and a tertiary amine group on the interior of the dendrimer. A dendrimer with buffering capacity demonstrates a proton-sponge result that enables escape from endosomes; it is also essential for conveying cationic nanocarrier into tumor cells. As a result, dendrimer functionalization and complete coverage of positive charges by PEG chain or acetyl group reduce dendrimer buffering ability [125]. The positive charges on the exterior of dendrimers are used not only for cationic dendrimer interaction with oligonucleotides but also for interaction between the cell membrane and dendrimer, which spearheads the internalization of dendrimer into tumor cells via cell membrane disruption [125]. The table below summarizes how transfection efficiency is improved by functionalizing dendrimers with various moieties.

#### 4.3.1. Transfection via Functionalization with Lipids

Due to the hydrophobic nature of the cell membrane and intracellular vesicles, functionalization of dendrimers using lipids leads to high cellular uptake and augmented endosomal escape. Fusogenic lipids, such as cholesterol and fatty acids, improve the cellular penetration of nanosystems in cancer cells [126]. PAMAM dendrimer linked to PEG and 1,2-dioleoyl-sn-glycero-3-phosphoethanolamine (G4-D-PEG-2000-DOPE) PAMAM dendrimers were used to condense small interfering RNA (siRNA), polyethylene glycol acted as a shield to protect siRNA from enzymatic deterioration, and DOPE enhanced cell membrane penetration by interacting with lipid molecules on the plasma membrane. They also created a combined micellar structure by mixing G4-D-PEG-DOPE micelles with PEG-5000-phosphoethanolamine (PE) micelles. When compared with plain PAMAM dendrimer, G4-D-PEG-2000-DOPE produced a significant increase in cellular uptake of siRNA and a notable downregulation. High micellar efficiency, drug loading, and cellular penetration led to high transfection and siRNA accumulation at tumor site [127].

#### 4.3.2. Transfection via Functionalization with Amino Acids

To reduce toxicity of dendrimers, anionic, cationic, and neutral types of amino acids are conjugated onto the surface of dendrimers, hence increasing transfection effectiveness [128]. Dendrimers with arginine- and lysine-functionalized surfaces have a higher positive charge density on their surfaces, which leads to more DNA condensation and cell membrane interaction [124]. The inclusion of phenylalanine and leucine amino acids (hydrophobic) tailors the hydrophobicity of the dendrimer surface, which is required for endocytosis. A study’s goal was to improve dendrimer transfection efficiency as a gene delivery vector in vivo and in vitro. Peptide-functionalized G5.0 resulted in efficient composite formation with plasmid DNA (pDNA) and shielding of pDNA from nuclease degradation. Furthermore, when compared with G6.0, it demonstrated high transfection effectiveness. The generation safety and transfection relation showed that peptide–dendrimer with optimum generation and molecular weight might be a good carrier for safe and effective gene delivery [129].

#### 4.3.3. Transfection via Fluorination of Dendrimers

Fluorinated dendrimers are highly efficient gene vectors, and fluorine internalization on drug molecules increases uptake, metabolic stability, and protein binding affinity. Fluorinated dendrimers thereby improve transfection effectiveness due to improved cellular uptake, stability in serum, and endosomal escape. With the proper sizes and zeta-potentials, these materials exhibit substantial DNA condensation and shielding ability. Fluorination of dendrimers upsurges the therapeutic effects and pharmacokinetics of several antitumor agents. Wang et al. created a fluorinated dendrimer in order to improve gene transfection efficiency. They tested the fabricated complex on various cancer cell lines and discovered that the fluorinated dendrimer achieved greater than ninety percent gene transfection in HeLa cells and HEK293 cells [130,131,132,133].

#### 4.3.4. Transfection Efficacy via Functionalization with Other Moieties

Increasing transfection efficiency can be accomplished through a variety of strategies, including the conjugation of cationic moieties (oligoamine, tertiary amine, quaternary ammonium, imidazolium, and guanidium) on the dendrimer surface, which greatly enhances the charge density on the dendrimer. CD (Cyclodextrin) is a cyclic saccharide with six (α), seven (β), and eight (γ) glycoprano units. They are typically used as a gene and drug delivery enhancer, protecting oligonucleotides from nuclease and facilitating endosomal escape of the pDNA complex after cellular uptake [124]. Qiu et al. created a gene delivery carrier based on G5.0 amine-terminated PAMAM dendrimer. They entrapped gold nanoparticles into dendrimers (AuDENPs) and coated their surfaces with β-CD to decrease toxicity and increase transfection efficiency. Then, they compacted pDNA encoding luciferase (LUC) and amplified green fluorescent protein (EGFP) on the dendrimer surface. On 293T cells, in vitro evaluation of the β-CDPAMAM dendrimer-AuNPs/pDNA complex revealed improved gene delivery compared with AuDENPs without the β-CD dendrimer and effective pDNA compaction [134].

### 4.4. Stimuli-Responsive Dendrimers

Stimuli-responsive nanocarriers reduce the toxic effect of chemotherapeutic agents in tumor therapy by sensing environmental changes in biological conditions. Controlled release and ease of application via a toggle switch system are advantages of stimuli-sensitive DDS, which improves release behavior of drug and its antitumor action due to the kind of stimuli and precise release location. Stimuli-sensitive carriers can be responsive to endogenic and exogenic stimuli, such as low pH, high glutathione concentrations, overexpression of specific enzymes, temperature, and light. As a result, stimuli-responsive DDS improves therapeutic efficiency while decreasing side effects [124].

#### 4.4.1. pH-Sensitive Dendrimers

Cancer cells obtain vitality by converting glucose molecules to lactic acid and accumulating lactic acid in cancerous tissue, which slightly reduces the pH of the tumor’s intracellular environment [135]. pH-sensitive DDS can be suitable for exhibiting on–off drug release behavior in tumor cells with no stimulation in normal conditions. Plain dendrimers with no functional groups on their exterior or interior can be counted as pH-responsive dendrimers due to ionizable functional groups such as amine and carboxylic acid on the exterior or core of dendrimers that can receive or give out protons depending on pH variations [136]. The principle of pH-sensitive release regulated by amine or carboxylic molecules is premised on amphiphilicity change caused by dendrimer charge change [137]. Thus, protonation of tertiary amine at low pH reduces the inner hydrophobicity of dendrimer and facilitates drug molecule release in the tumor site. The attachment of acid-label linkers allows for the fabrication of pH-sensitive dendrimers. When the pH in the tissue is neutral or alkaline, pH-sensitive linkers remain stable, but they hydrolyze due to acidic pH. Hydrazone, acetal, oxime, b-carboxylic acid, and amine groups are pH-sensitive linkers commonly utilized in conjugating drugs on the surface of dendrimer [119].

#### 4.4.2. Enzyme-Responsive Dendrimers

The alteration of certain enzymes expressed in various pathological states, such as cancer, develops an enzyme-sensitive structure as an efficient strategy for cancer therapy centered on sustained and regulated delivery behavior [137]. The most significant advantage of enzyme-responsive release is the absence of the necessity for exterior stimuli to cause responsive release. Hyaluronidase, esterase, matrix metalloproteinases, and cathepsin B are forms of enzymes that are highly expressed in cancer tissue and are thought to be suitable activators [138,139]. The most common functional groups that could be incorporated into dendrimers to functionalize enzyme-sensitive vectors are the ester moiety and short peptide sequences. A notable controlled drug release carrier for cancer therapy is an enzyme-responsive dendrimer [140,141].

#### 4.4.3. Redox-Responsive Dendrimers

Redox-sensitive carriers should have the ability to shield bioactives in cell membranes and in plasma. Still, due to their responsiveness to possible intracellular reduction, loaded bioactives are released into tumor cells [142,143]. Disulfide bonds are commonly utilized as redox-responsive inducers in cancer therapy because they are stable out of the cells and in circulating blood but unstable when transported into cells [139]. Dendrimers with disulfide bonds exhibit redox-responsive controlled release behavior via several major approaches: (i) conjugation of disulfide bond onto the dendrimer surface, (ii) disulfide bond between dendrimer and protective ligands such as PEG, (iii) disulfide bonds being dendrimer core and monomer, (iv) disulfide bond formation of shield on dendrimer surface [141]. A significant barrier to dendrimer transfection potential is their toxic effects related to generation—that is, a low-generation dendrimer results in low toxic effects and transfection efficiency and the opposite is true. This conundrum could be solved by cross-linking low-generation dendrimers to disulfide bonds.

#### 4.4.4. Thermoresponsive Dendrimers

As malignant tumors have a slightly higher temperature than normal tissue, exploiting this trait may be a suitable approach for achieving controlled release of drug [138]. It is vital to observe that the release of a selective encapsulated payload into cancer tissue necessitates the use of an external heating mechanism [144]. Direct attachment of thermoresponsive polymers on the dendrimer facet and alteration of structural elements of thermosensitive polymers on dendrimer facet, core, and internal branches are two methods used to impart thermoresponsive dendrimer [136,141]. Temperature-dependent polymers experience a shift in water solubility as the temperature changes, which is used to provide a thermoresponsive delivery system. Water solubility changes are due to a lower critical solution temperature (LCST), also known as the cloud point. When the temperature drops below the cloud point, polymers become soluble; when the temperature rises above the cloud point, polymers aggregate and become insoluble. Poly(N-isopropylacrylamide) (PNIPAM) is a well-known thermosensitive polymer having a cloud point of 31–32 °C. Due to a change in cloud point, this polymer undergoes a phase transition from hydrophilicity to hydrophobicity. Dendrimer amphiphilicity is significantly reduced at temperatures above the cloud point [145]. This characteristic can be utilized to create temperature-sensitive dendrimers for the delivery of drugs. Increased incubation time above the cloud point, on the other hand, can enhance cellular uptake of thermosensitive dendrimer [138]. Another polymer used to make heat-sensitive dendrimers is poly (N-vinyl isobutyramide). Polymer structural units are also exploited as thermosensitive moieties for functionalization of dendrimers and translation to temperature-responsive dendrimers [141]. It is worth noting that amino acids and peptides (e.g., collagen, phenylalanine) can endure phase transition in dendrimers and are classified as thermoresponsive peptides. High-generation dendrimers exhibit low cloud point [141].

#### 4.4.5. Photoresponsive Dendrimers

The photoresponsive dendrimer structure is based on two pathways: (i) irradiation at one time and (ii) repetitive on and off discharge of encapsulated or adjoined elements by irradiation with a specified wavelength [137]. Based on photo release, three wavelengths are used in cancer treatment: near-infrared region (NIR) 650–900 nm, visible light 430 nm, and ultraviolet (UV) 100–400 nm. Ultraviolet and visible light are not appropriate for in vivo use due to their inability to penetrate deep tissue and toxicity, whereas NIR light can. Furthermore, NIR does not harm surrounding tissue because of its low phototoxicity [141,142]. In photoresponsive delivery systems, the photochemical internalization effect is accountable for the easy delivery of drugs and genes into the cytoplasm. For example, when a dendrimer–porphyrin core is taken up by an endosome and exposed to UV light, the endosome membrane is disrupted, resulting in gene/drug discharge into the cytosol.

Furthermore, porphyrin-core lysin dendrimer compounded with DNA and adjoined to cationic peptide prior to laser irradiation increased transfection efficiency [141]. Light-irradiation-directed drug is conjugated to dendrimer via adjoining the photosensitizer with dendrimer, linked by hydrazone bond. When a photosensitizer is activated by light, it generates ROS in the endosome, which leads to endosomal membrane disruption and drug molecule escape [146]. Azobenzene and Ortho-nitro benzyl (ONB) are two notable linkers that can photoisomerize when exposed to UV light and are commonly utilized to functionalize light-sensitive dendrimers [142].

### 4.5. Multifunctional Dendrimers

Sometimes, only one functionalization is insufficient to curb all setbacks, necessitating multifunctionalization [124]. PEGylation, for example, increases the stability and lengthens the circulation period of a nanocarrier, resulting in nanocarrier buildup at the tumor site. Furthermore, PEG chains have a decreased cellular uptake due to an elevated hydrophilic corona and a high steric hindrance. As a result, dual or more functionalization of dendrimers with different targeting molecules may be able to overcome this drawback in cancer DDS. Biswas et al. developed an interiorly quaternized and externally acetylated PAMAM dendrimer (G4.0) for targeting siRNA within tumor cells. Acetyl-dendrimer was less toxic than intact dendrimer, and its cationic internals made an effective compact with siRNA to protect it from environmental deterioration [127]. Zhu et al. developed a multifunctional nanocarrier that can be used for imaging and cancer treatment. They coupled PEGylated-tocopheryl succinate (α-TOS) and folic acid-PEG in PAMAM dendrimer G5.0, which is amine-terminated, then adjoined FITC to the surface of dendrimer and trapped Au nanoparticles (AuNPs) in the dendrimer inner cavity (AuDENPs). The remaining amine groups of the dendrimer were then modified with acetylation to form an unbiased, multifunctional Au-TOS-FA-DENPs structure with higher therapeutic effectiveness and more stability than free α-TOS. The multifunctional structure was directed at FR overexpressed cells resulting in effective imaging using trapped AuNPs [147,148].

## 5. Silica-Based Nanoparticulate Drug Delivery System

Colloidal silica nanoparticles are amorphous materials with generally spherical shapes [149]. The particle sizes can be varied, and the surface chemistry can be easily modified to target various applications [150,151]. Silica nanoparticles are absorbent and abrasive in their impermeable form, whereas mesoporous silica nanoparticles (MSNPs) have applications in nano drug delivery [149,152]. In this respect, MSNPs with a dimension of less than 500 nm have remarkable capabilities for multiple drug delivery applications [153]. MSNPs are a hollow nanocapsular framework that allows drug molecules to be conjugated via open functional groups [154]. They are naturally firm and have a flexible symmetry on the surface that can be further adjusted through chemical functionalization. MSNs could be synthesized using four different procedures, which have been reported in the literature: template-directed method [155], sol–gel technique [156], microwave-assisted method [157], and chemical etching method [158]. Furthermore, MSNPs enable improved surface functionalization and porosity, allowing molecular cargos to be hosted without disrupting the silica structure [159,160]. Simple silane chemistry allows for the surface functionalization of MSNPs with a wide range of functional groups, making them uniquely suited for various therapeutic applications [161,162].

The US Food and Drug Administration (USFDA) has recognized mesoporous silica-based carriers as “Generally Recognized as Safe (GRAS)”, making them the most acceptable nanosystems for clinical application as prospective therapies for the treatment of diseases in the contemporary era [163]. Adding molecular functionality onto the surfaces of silica platforms can drastically change the features of the acquired molecule, which are crucial for molecular identification with the payload. The insertion of organic moieties at odd positions is recommended to perfect the surface of the particles [164,165]. Various methods have been proposed to obtain the functionalization of silica materials, including periodic mesoporous organo-silica formation and metal-organic reagents. However, postsynthetic grafting and co-condensation are the two most extensively utilized techniques for functionalization [166]. These methods open new avenues for designing and fabricating highly functionalized MSNPs for regulated drug release and cargo molecule protection from enzymatic degradation [167]. The drug loading and delivery possibilities of mesoporous silica particles can also be influenced by electrostatic interactions between drug moieties and the surface of silica [168]. The application of functional changes to MSNPs can aid in the controllable delivery of payloads at the targeted sites without premature delivery while in blood circulation [169]. This is capable of notably reducing the drug molecules’ side effects and improving their therapeutic potential [170].

In this study, biochemical modification of silica-based nanoparticles was demonstrated. Enzymes and biologically compatible chemical reagents were used to modify the surfaces of pure and dye-doped silica nanoparticles, allowing them to operate as biosensors and biomarkers. After the Stober synthesis, pure silica NPs were synthesized, salinized, and functionalized with an enzyme. To undergo biochemical modification, salinized NPs were exposed to lauroyl chloride in dry tetrahydrofuran. The experiment’s findings showed that silica nanoparticles provide an excellent biocompatible solid substrate for enzyme immobilization—the immobilized enzyme molecules on the nanoparticle surface displayed great enzymatic activity in their respective enzymatic activities. The potential of utilizing nanoparticles for biosensing and biomarking applications in cancer therapy was proven by the biochemical functionalization of nanoparticle surfaces [171].

### 5.1. Temperature-Sensitive MSN

Poly(N-isopropylacrylamide) (PNIPAM), poly(-caprolactine) (PCL), and short peptides are some of the well-known temperature-sensitive polymers used in the formulation of temperature-sensitive MSN. PNIPAM can be synthesized separately before conjugating to MSNs, or it can be prepared directly by copolymerizing N-isopropylacrylamide monomers (NIPAM) on the surface of MSNs with modified methacrylate or vinyl groups. ATRP initiator could be conjugated to MSNs prior to initiating NIPAM ATRP to directly produce PNIPAM from the surface of MSNs. PNIPAM’s temperature change feature can be utilized to regulate the delivery of bioactives from MSN pores either by exposing the pores to the neighboring environment for the release of bioactive moieties (MSN with large pore sizes and short conjugated polymer chain lengths) or by entrapment of cargos in the cavities of MSNs to allow for sustained release with minimized initial burst (smaller pores MSNs and lengthy chains of polymers conjugated). Other moieties, such as poly (-caprolactine) and short peptides, have also been discovered as temperature-responsive molecules that are conjugated on the MSNs surface, acting as a “gateway”; these molecules are adjoined to the surface of MSNs via EDC chemistry and the “click” reaction. Temperature-sensitive MSNs can provide localized, controlled, and sustained delivery systems that prevent systemic cytotoxicity [172,173,174,175,176,177,178,179].

### 5.2. Light-Sensitive MSN

Derivatives of orthonitrobenzyl (ONB), derivatives of coumarin, thymine, derivatives of azobenzene (Azo), and spiropyran are among the many moieties used in light-sensitive MSN production. Bioactive molecules are loaded into MSN pores, and the ONB-contained capping fragments will be covalently or electrostatically conjugated to the surface of the MSNs to avoid payload release. When these formulations are exposed to Ultraviolet light, photodegradation of the ONB group results in the release of capping fragments and, eventually, the payload. Coumarin-derived molecules can also degrade when exposed to UV or NIR light. Coumarin was adjoined to the surface of MSNs using urethane linkage; then, the cargo was loaded and capped with cyclodextrin to avoid diffusion; 2.5 h later with no UV light, no released payload was detected, whereas >40% and 70% of cargo were released after 2.5 and 7.5 h, respectively, after switching on ultraviolet. Cyclodextrin can also be functionalized onto MSNs before encapsulation of drugs and coating with Azo-containing polymers. When the light (365 nm or 625 nm) was shone onto MSNs, the trans to cis conversion caused the Azo-Cyclodextrin inter-reaction to break, allowing the discharge of capping molecules and, subsequently, the release of loaded bioactives. Spiropyran is a hydrophobic photochromatic molecule that, when exposed to ultraviolet light, transforms to a hydrophilic positively charged merocyanine but returns to the original spiropyran-form when exposed to visible light; however, the merocyanine form is unstable without light, which may result in leakage of payload. Therefore, the spiropyran form is maintained and used for storing and circulation of payload. In a study, before loading the bioactive molecules, hydrophobic compounds, perfluorodecyltriethoxysilane, and spiropyran were used to functionalize the surfaces of MSNs. The hydrophobicity of spiropyran and perfluorodecyl creates a thin hydrophobic barrier that hinders MSN payloads release. Under UV light, spiropyran was transformed to the hydrophilic merocyanine form, which breached the hydrophobic barrier and wetted the MSNs surface, enabling laden molecules to be discharged. It provides numerous benefits, including ease of control, extremely high spatial and temporal precision, remote control, and a wide range of wavelength and intensity [180,181,182,183,184,185,186,187,188,189,190,191,192,193,194,195,196,197,198,199,200,201,202,203,204,205,206,207].

### 5.3. Magnetic-Field-Sensitive MSN

Superparamagnetic ferric oxide nanoparticles (M-Fe-NPs) examples: Fe_3_O_4_ and γ-Fe_2_O_3_ are the most popular sources of magnetics that can be incorporated to produce magnetic-field-sensitive MSN. Carbon-encapsulated ferric magnetic colloidal nanoparticles, Gd ions, and MnOx are other examples of magnetic moieties. The fabrication and release process of cargo from functionalized MSNs with magnetic response includes MSNs with magnetic NPs in the inner core, amino stalks connected to the MSN’s surface, drug loading then capping with Cucurbit (6) via host–guest interaction, and application of oscillating EMF to elicit hyperthermia effect for discharge of the capping fragments and loaded drug. Under the influence of an external magnetic field (EMF), M-Fe-NPs exhibit hyperthermia. M-Fe-NP hyperthermia has been used to solely regulate payload release or to trigger the detachment of capping fragments to regulate the release of bioactive molecules, controlling site-targeting, and MRI [181,208,209,210,211,212].

### 5.4. Ultrasound-Sensitive MSN

Ultrasound can be incorporated by introducing some of these moieties (Liquid Perfluorohexane (PFH), Hydrophobic floured alkyl or alkyl; Pluronic F127 solution, 4,4′-azobis (4-cyanovaleric acid) (ABCVA)). The encapsulation of the PFH bubble generator improves the resolution of the ultrasound image, making it easier to focus the therapeutic site within the targeted tumor tissue by using gas bubbles to entrap drug-loaded MSNs. Links between MSNs and capping molecules: PEG was utilized to partly cap topotecan (TPC)-loaded amino-functionalized MSNs through 4,4′-azobis (4-cyanova leric acid) (ACVA) thermodegradable linkers to block untimely drug diffusion and modulate the surface charged of MSNs. At 70 °C or after exposure to Ultrasound radiation, ACVA decomposes to produce two free radicals because of the localized hyperthermia effects, which initiate the cleavage of links between MSNs and PEG, followed by the departing of PEG for TOP release and the exposure of the amino group to enhance the positive surface charge of MSNs for improved cell internalization. EDC chemistry: Using EDC chemistry, a copolymer poly (2-(2-methoxyethoxy) ethyl methacrylate)-co-PTHPMA (p (MEO2MA-co-THPMA) was conjugated to the surface of MSNs for ultrasound-stimulated DOX release. Following ultrasound irradiation, the hydrolysis of PTHPMA increased the polymer’s hydrophobicity due to the formation of –COOH groups, which resulted in the formation of a coil-like structure that allowed for the release of DOX. Ultrasound can cause hyperthermia and cavitation in functionalized MSNs, causing them to respond to cofunctionalized ligands and/or deliver loaded payloads [213,214,215,216,217].

### 5.5. Electric-Field-Sensitive MSN

Bipolar Molecules example: Bipolar 4-(3-cyanophenyl) butylene Bipolar 4-(3-cyanophenyl) butylene molecules were functionalized into the core surface of MSN pores and served as nano impellers under the application of an external electric field to regulate the release of loaded IBU. Controlling the frequency of the exterior electric field could control the shifting rate of the bipolar molecules, allowing the release rate of loaded IBU to be well-tailored. The discharge of bioactives from a hybrid MSN/hydrogel system can also be triggered by a static electric field. Using an electrode-position technique and chitosan solution containing IBU-loaded MSNS, the hybrid MSNs/hydrogel system was glazed on the surface with a titanium plate. Without a static electric field, below 30% of the drug was released from the system, whereas nearly 100% of IBU was released after 3 h of trigger at the cathodic voltage of 5.0 V. Electric field has also been used to tailor the release behavior of encapsulated biomolecules from MSNs [218,219,220,221].

### 5.6. pH-Sensitive MSN

Functionalization molecules containing carboxylic acid (COOH) amine examples: poly(3-(3-methacrylamidopropyl-(dimethyl)-ammonio) propane-1-sulfonate)-grafted polydopamine (PDA), 3-aminopropyltriethoxysilane, poly(2-(dimethylamino) ethyl acrylate), poly(2-(pentamethyleneimino) ethyl methacrylate), and polylysine-grafted polyethylenimine copolymers are the simplest approaches to introduce pH sensitivity to MSNs. Acid-labile functional groups such as acetal linker, amide linker, boronate linkage, Schiff base linkage, hydrazone bond, and cis-aconitic acid derivatives are used as well. Metal minerals, oxides and salts, and lipids and lipid-conjugated polymers are not excluded. Covalent functionalization or physical coating on the surface of MSN to deliver electrostatic interaction with cationic bioactive molecules, or to act as capping pieces to prevent loading bioactive molecules from diffusing prematurely. The functionalized MSNs encapsulated with cargos are stable for long-term circulation at physiological pH without causing systemic cytotoxicity. The system is activated when the pH decreases, for example, in the tumor microenvironment or in the cytoplasm, due to the breakdown of COOH-containing ligands, which leads to the opening of pores for discharging the payloads. On the other hand, amine-containing ligands are conjugated to MSNs in order to cause electrostatic interactions with anionic bioactive components or to operate as regulation layers to block payloads in pores. The breakdown of acid-labile linkers in the presence of acidic pH resulted in the detachment of capping fragments and the release of encapsulated bioactive molecules. Furthermore, metal minerals, oxides, and salts can be utilized in functionalization of MSNs in order to increase their pH-response characteristics due to metal dissolving at low pH. Functionalization can occur during MSN fabrication or as a capping procedure after the cargo has been loaded. Lipids or their conjugated polymers form a pH-responsive capping layer. The disruption of the lipid membrane caused a storm increase in payload when exposed to pH 5.0. Ph sensitive MSN minimizes systemic cytotoxicity, controlls release of payload, and protects the diffusion of the loaded biomolecules for long-term circulation and stability under physiological pH [218,222,223,224,225,226,227,228,229,230].

### 5.7. Redox-Sensitive MSN

Redox molecules include glutathione (GSH), dithiothreitol (DTT), and tris(2-carboxyethyl) phosphine. The incorporation of functionalization ligands into MSNs through redox-degradable disulfide bonds has been applied to deliver chemo-payloads intracellularly. From one of the simplest methods, bis-(N-3(triethoxysilyl) propyl 3-carboxamide-4-hydroxy phenyl) disulfide was observed on the surface of DOX-loaded MSNs to act as a redox-responsive intermediary. The following are common approaches for ligand surface modification on MSNs mostly through disulfide bonds: (1) surface modification of MSNs with thiol groups or pyridine (Pyr) via disulfide bond accompanied by capping via disulfide exchange or host–guest interaction with Pyr; (2) surface modification of MSNs with –COOH or primary amino groups via disulfide bond preceded by conjugation of capping pieces; (3) chemical or physical coat on the surface of MSNs with a polymer containing a disulfide bond to serve as intermediaries. Saline direct deposit and click reaction methods are also used to enhance the bioactivity and minimize the systemic cytotoxicity [231,232,233,234,235,236,237,238,239,240].

### 5.8. ROS-Sensitive MSN

Moieties such as phenylboronic acid, arylboronate pinacol ester, and AgNPs are stable in normal biofluid but are degraded in ROS such as H_2_O_2_. To avoid cargo diffusion, bioactives were loaded into phenylboronic-acid-conjugated MSNs before being capped with adjacent diol groups, such as human IgG and ß-D-glucose-AuNPs. Upon exposure to H_2_O_2_ in the specific microenvironment, the phenylborate ester bonds deteriorated to make phenol groups; thus, the covering moieties disconnected from the surface of MSNs, resulting in an increase in payload release rate and a controlled release [241,242,243,244].

### 5.9. Enzyme-Sensitive MSN

Oligo DNA containing telomere repeat complementary sequences CCCTAA, gelatin or peptides containing PLGVR or PVGLIG sequences (which is degraded under exposure to MMP-2), and Cytosine phosphodiester-guanine oligodeoxynucleotide (CpG ODN) have all been reported to be used as moieties for enzyme-sensitive MSN. To avoid diffusion and preserve the loaded fluorescein or DOX through circulation, oligo DNA having telomere repeat complementary sequences CCCTAA was coated on the surface of MSNs. Fluorescein or DOX were storm released for imaging or chemotherapy in the presence of telomerase enzyme in the cell cytoplasm. Many different enzymes have been found to be highly expressed in cancer cells and tumor tissues, including matrix metalloproteinase 2 (MMP-2), Cathepsin B, and hyaluronidase (HAase). It has also been reported that a specific colonic enzyme causes the release of 5-FU chemotherapeutics from functionalized MSNs. Other functionalized moieties, including cytosine phosphodiester-guanine oligodeoxynucleotide (CpG ODN), Ag-stabilized triplex DNA formed by complexing a double-strand DNA NF-kB p50 transcription factor (TF) and an Ag single-strand DNA, and poly (lysine-dopamine), have been shown to dissociate when exposed to deoxyribonuclease I, TF, and Pepsin, respectively, for regulated release of cargos. Using these enzymes to control the intracellular release of chemotherapeutics at the local tumor site for cancer treatment can reduce the amount of drug used, improve treatment efficacy, and reduce systemic cytotoxicity. This can also result in a mild intracellular condition, high selectivity, and low side effects [245,246,247,248,249,250,251,252,253].

### 5.10. Functionalization of MSN for Other Responses

ATP is used as a sensor to regulate the delivery of molecules from MSN-based nanosystems because its expression goes up in some cells or biological mechanisms, particularly in tumor cells and neural stem cells [254]. The ATP-aptamer have a distinct characteristic trait and chemical framework that allows it to attach to ATP molecules, which is often used to regulate the release of bioactive substances from MSNs [255]. To avoid cargo diffusion in the lack of ATP, adenosine was functionalized on the facet of MSNs for topping with ATP-aptamer modified on AuNPs. When ATP was launched into the release media, the cargo was discharged as an outcome of the competitive affinity of free ATP with ATP-aptamer coupled with AuNPs, preceding the dissociating of topped AuNPs. The ATP-aptamer was also functionalized just on the facet of MSNs by means of eight prolonged bases that are able to create a duplex system with the aptamer’s first eight bases to function as an entrance to close the pore and effectively block payload diffusion [256]. When ATP is present, the duplex system transforms into a hairpin system, allowing the cargo to exit through the pore [257]. Metal divalent ion compounds are utilized as capping fragments for ATP-trigger cargo discharge in the presence of ATP since ATP has a strong affinity for metal divalent ions such as Cu^2+^ and Zn^2+^ [258]. It is also worth mentioning that MSN can be functionalized to be glucose-responsive. Diabetes patients have remarkably high blood sugar levels, a vital signal to regulate insulin release—a protein substance for diabetic therapy [259,260,261].

### 5.11. Multiple-Stimuli MSN

The tissue microenvironment contains the majority of the biosignals that control the reaction of functionalized MSNs. Cancer tissues, for example, have a lower pH, and tumor cells are high in redox elements, enzymes, and targeting receptors. Moreover, outside signals such as light, temperature, magnetic fields, and electricity are ready and simple to regulate [262,263,264]. As a result, functionalizing MSNs with multiple ligands can increase the synergistic effect in controlled therapeutic delivery alongside reducing the systemic cytotoxicity of toxic biomolecules [265]. Temperature [172,173,174,175,176,177,178,179], pH [218,222,223,224,225,226,227,228,229,230], magnetic [181,208,209,210,211,212] and electric fields [218,219,220,221], glucose [259,260], enzyme [245,246], [245,247,248,249,250,251,252,253], ATP, and ultrasound [214,215,216,217] have all been reported as combination ligands. PH-sensitive ligands are cofunctionalized with ligands that respond to temperature, magnetic field, light, ATP, glucose, enzyme, ultrasound, and ROS to improve delivery effectiveness and therapeutic outcome [266]. Numerous amalgamations have been revealed to deliver versatile tools for controlling the release of bioactives [267]. These functionalized MSN systems were deliberated upon in the preceding definite sections.

### 5.12. Functionalization of MSN for Cell Targeting

Toxic biomolecules, particularly chemotherapeutics, may be delivered systemically by MSN-based nanocarriers, causing unintended cytotoxicity to healthy cells. MSNs are functionalized with precise cell-targeting ligands to focus the nanosystem to the specified cells because of the presence of cell-specific receptors and reduce the systemic cytotoxic effect [268]. For instance, DOX-loaded TAT-modified MSNs [269] notably augmented intracellular and intranuclear DOX concentrations in MDR MCF-7/ADR tumor cells at a considerably increased level compared with the free DOX. TAT-peptide with YGRKKRRQRRR sequence can attach to tumor cells accessible importin and receptors, then aim at the cell nuclear pore complexes to enter the nuclei [270]. The AS141-aptamer sequence has a high specific recognition ability with nucleolin on the surface of tumor cells, and it has been commonly utilized in tumor cell targeting to deliver chemotherapeutics. Cancer cells have various types of receptors (folate receptor alpha, CD44 receptor, asialoglycoprotein receptor (ASGPR), etc.) on their surfaces that can be used for targeted delivery of nanosystems [271]. Other moieties aimed at cancer cell surface receptors, such as RGD-containing peptides targeting integrin receptors, Tf peptides targeting Tf receptors (TfR1 or CD71), and phenylboronic acid that aims at sialic acid residues, have been revealed to improve cancer cell targeted delivery of functionalized MSN nanocarriers significantly [272].

## 6. Limitations of NPDDSs

Some challenges and limitations still faced by both functionalized and nonfunctionalized nano drug delivery systems include the following: (1) Use of proper storage techniques. For example, liposomes should be stored in the refrigerator and not the freezing compartment of the same unit. Storage in the freezer will cause the formation of ice crystals, which may break the liposome’s phospholipid bilayers [273]. (2) Choice of the most convenient route of administration. The most common route of administration is the oral route, which in most cases is more convenient; however, most of the current commercially available formulations for cancer therapy are administered by the intravenous route [274]. (3) Development of NPDDSs with appropriate morphological properties such as size, shape, and charge for enhanced cellular uptake, and lower ligand–receptor interactions. Shapes such as spheres, rods, and cylinders have been shown to enhance cellular uptake. Positively charged liposomes are rapidly absorbed by the cells due to electrostatic attraction within the liposomes and the negatively charged cell membrane [275,276]. Optimal ligand density on the surface of liposomes can increase uptake in cancer cells. Conversely, exceeding the optimum saturation of ligands will result in aggregation. Challenges such as the features of targeted cells, functionalization leading to enhanced detection by the immune system, interaction between ligands and serum proteins, and stability of formulations to obtain increased cytotoxic effects need to be addressed [277]. (4) Drug leakage after administration has also been reported, making the scale-up of functionalization techniques from small scale laboratory preparation to large-scale industrial application very challenging [278]. Numerous existing nanoparticle production techniques are only best suited for lab-scale production. Low-energy-input techniques for large-scale fabrication of nanoparticles must be devised. Furthermore, limited by the size and large specific surface area of nanoparticles, dry forms of nanoparticles are easily aggregated, making them hard to deal with. Early detection of cancer cells to reduce patient recovery time and drug side effects to healthy cells is the need of the hour. More investigation is necessary on cancer recurrence, multidrug resistance, and the challenge of penetrating two barriers—the BBB and the tumoral barrier [279,280,281,282,283]. In this regard, improvements to existing methodologies or the implementation of innovative methods may be beneficial.

## 7. Conclusions

Functionalizing nanoparticulate drug delivery systems (see in Figure 4) yields the ultimate qualities needed for a specific application. In the realm of nanopharmaceuticals, the significant goals of functionalization are to make nanoparticles’ immune systems unidentifiable, lead them to the appropriate spot, and generate a high number of therapeutically active treatment solutions. Significant research has been conducted to assess the properties of functionalized nanoparticles in terms of their suitability for drug delivery. So far, the functionalization of nanoparticles has been of more benefit than harm in the nanopharmaceutical field. Even though the functionalization of nanoparticles is not new, there are still several researches and experiments that have been done and are still ongoing to improve existing synthesis methods and develop new and better strategies. Multifunctionalization is trending in this field of study and may be of more significant benefit. In conclusion, the unique properties of the various functionalized nanoparticles should be identified, such as their physicochemical properties and their ability as carriers of anticancer agents, to widen their applications.

## Figures and Tables

**Figure 1 pharmaceutics-14-01113-f001:**
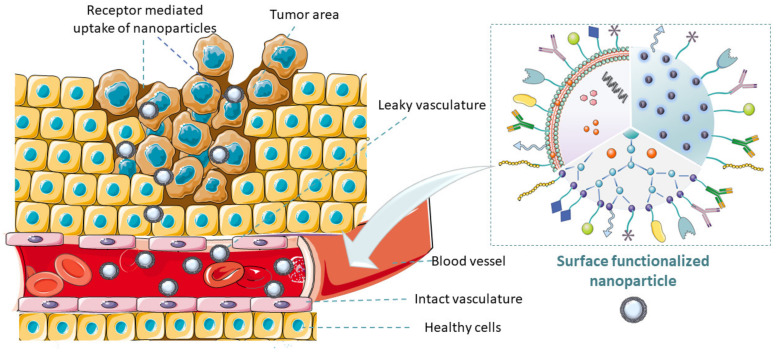
Graphical representation of the abstract. Illustration of tumor microenvironment and how surface-functionalized nanocarriers containing antitumor drugs actively target tumor cells.

**Figure 2 pharmaceutics-14-01113-f002:**
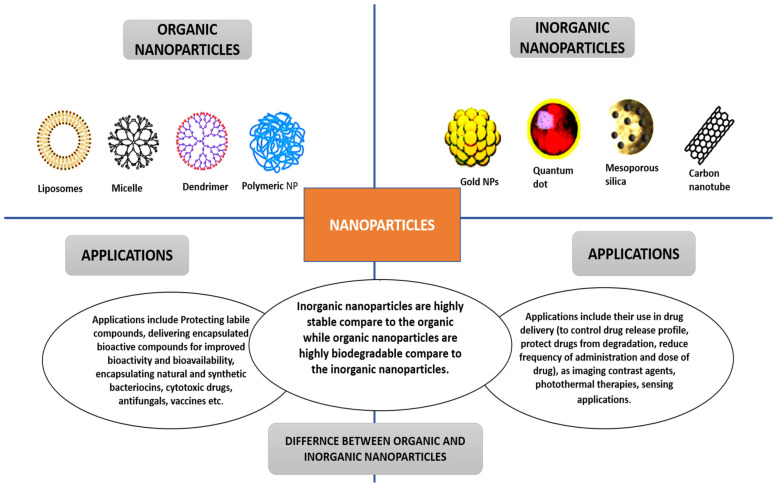
Nanoparticles’ classifications, applications, and differences.

**Figure 3 pharmaceutics-14-01113-f003:**
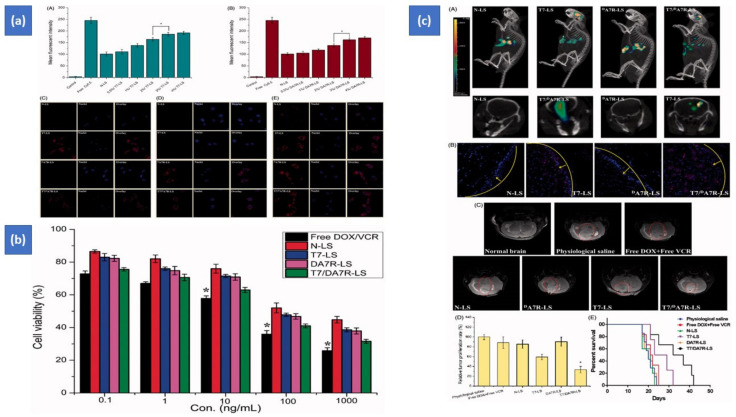
Targeted liposomes having two peptides (TfR- and VEGFR2-specified peptides) and two antitumor agents (doxorubicin and vincristine). (**a**) Cellular uptake (After 2 h at 37 °C, cellular uptake of Cy5.5-loaded liposomes of varying densities of T7 (**A**) and DA7R (**B**) in C6 cells. The cells’ auto-fluorescence was used as the control. Cellular uptake of varied Cy5.5 loaded lyposomes by bEND.3 cells (**C**), HUVECs (**D**), and C6 (**E**) cells); (**b**) cytotoxicity study (The cytotoxic activity of free DOX + free VCR, as well as some liposomes containing DOX and VCR); (**c**) biodistribution study (The biodistribution of Cy5.5 in varied liposomes in mice with intracranial C6 glioma was ascertained using an IVIS® Spectrum-CT (**A**). A CLSM was used to show the allocation of Cy5.5 in the brains of mice with intracranial C6 glioma (**B**). 16 days after inoculation, MRI of physiologic and pathological brains (**C**). Glioma tumor cell division rate in the brain (**D**). Survival curves according to Kaplan–Meier (**E**). The yellow line represents the intracranial glioma margin, and the arrow represents the glioma cells. The red is Cy5.5, and the nuclei are stained with DAPI (blue). Effectiveness after treatment with different formulations at 1 mg/kg (DOX 0.8 mg/kg + VCR 0.2 mg/kg) on days 8, 10, 12, and 14 after inoculation.) This designed system could go through the blood–brain barrier and blood–tumor barrier, with enhanced cellular uptake and cytotoxicity [32]. Copyright 2017, Taylor & Francis Journals. * *p* < 0.05.

**Figure 4 pharmaceutics-14-01113-f004:**
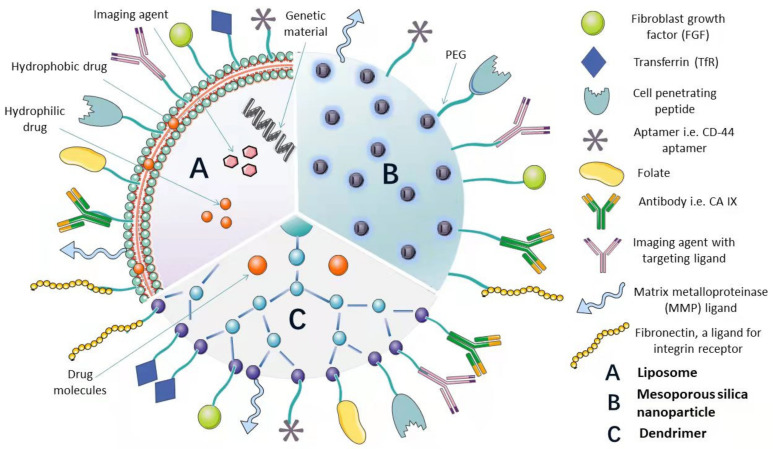
Illustration of the types of functionalization on the nanoparticulate drug delivery systems.

**Table 1 pharmaceutics-14-01113-t001:** Liposomes’ surface modifications with various moieties and their applications.

Type of Moiety	Application of Functionalized Liposome	Ref.
*Vitamins*		
Biotin	To target EGFR, Quantum dots were coupled to an epidermal growth factor ligand.	[44]
Vitamin A	Skin fibrosis is treated with a siRNA carrier. siRNA carrier to resolve liver cirrhosis	[45,46]
Folic acid	Macrophage targeting with ovarian carcinoma. Oligodeoxynucleotide targeting cancer cells.	[47,48]
*Carbohydrates*		
Glucose	Drug delivery for capillary endothelial cells in the brain.	[49]
Sucrose	Doxorubicin-loaded liposomes for cancer treatment	[50]
Lectins	Pulmonary drug delivery	[51]
*Antibody fragments*		
scfv	Trastuzumab–Liposomes for advanced breast cancer	[52]
Anti-CD 133 Mab	Bevacizumab-containing liposomes for glioblastoma	[53]
Anti-transferrin scFv antibody fragment	Plasmid DNA-carrying liposome for prostate cancer cell lines	[54]
*Aptamer*		
IL-4R⍺	Tumor growth inhibition through targeting the tumor microenvironment	[55]
xPSM-A9	To combat the expression of a membrane antigen (prostate specific) on prostate cancer cells.	[56]
Anti-CD44	Selectively targeting cancer cells	[57]

**Table 3 pharmaceutics-14-01113-t003:** Examples of stimuli responsive dendrimers are summarized.

Nanocarrier	Stimulus	Modifier	Drug/Disease	Ref.
mPEGylated dendrimer	Esterase enzyme	Succinate-linker	Paclitaxel/cancer	[118]
Degradable dendrimer	UV irradiation	o-Nitro benzyl	DNA	[119]
PAMAM dendrimer-PEG	Redox/Glutathione	Disulfide bond	Doxorubicin/lung cancer-A549-B12	[120]
PAMAM dendrimer-PEG-gold nanorod	pH stimuli	Hydrazine-linker	Doxorubicin/cervical cancer-Hela	[121]
PEGylated lysin peptide dendrimer	CathepsinB enzyme	GFLG (gly-phenylalanyl-leucyl-glycine)	Gemcitabine/breast cancer	[122]

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
