# Peer review of "Functionalization of Nanoparticulate Drug Delivery Systems and Its Influence in Cancer Therapy"

_pharmaceutics, 2022, doi:10.3390/pharmaceutics14051113_

Round 1

Reviewer 1 Report

The authors present a review manuscript describing the surface functionalization of various types (organic and inorganic) of nanocarriers in order to achieve an enhanced effect in cancer therapy. The review covers functional liposomes, dendrimers and mesoporous silica as nanocarriers for targeted drug delivery. Unfortunately, I have some serious concerns about the review in its present form and I think it should be carefully revised prior to be considered for publication.

  1. The reference list formatting has nothing in common with the instructions for manuscript preparation for this journal. Moreover, not giving the whole authors list, for example “E. Andreozzi et al.”, is not acceptable at all.
  2. When I tried to follow a reference, for example Ref. 157, which is cited several times, it turns out that in the reference list Ref. 157 is about something completely different from what is discussed in the text. Then, I found more and more wrongly cited references. Overall, it turns out that the content in the paper is not supported by appropriate references. Again, it is unacceptable to present such a messy literature list for peer review and it raises many concerns about the quality of the paper.
  3. The authors covered liposomes and dendrimers from the organic nanocarriers. However, the most widely used and versatile organic drug delivery systems are based on the amphiphilic block copolymer nanoparticles (micelles, polymersomes etc.). Unfortunately, this class of functional nanocarriers is skipped in the review manuscript. It would be nice if functional polymer drug delivery systems are covered as well since they are much more significant in terms of drug delivery compared to the exotic dendrimer macromolecules.
  4. I think that Tables 4 and 5 contain too much text in the columns and it would be better to convert them into plain text.
  5. Other minor points:
  • page 2: it should be “nanocarriers” instead of “nanoacciers”;
  • page 7: The abbreviation MAL (maleimide) should be explained after the first use;
  • page 13: there is a complete reference listed in the text with no need. It should be removed from the text;
  • page 15: the abbreviation NIPAAm is for the monomer N-isopropylacryamide and not for the corresponding polymer, which is PNIPAAm. The same is valid for 2-propyl acrylic acid – it is a monomer and not a polymer as it is written in the text;
  • the term “lower critical stability temperature” should be corrected to “lower critical solution temperature (LCST)”.

Author Response

Our response is in the PDF file below. Thank you

Reviewer 2 Report

The review is interesting and authors have made significant efforts to cover a broad range of topics. 

Author Response

Our response is in the PDF file attached. Thank you.

Reviewer 3 Report

My Referee Report is attached in pdf format.

Author Response

Our response is in the PDF file attached. Thank you

Reviewer 4 Report

An interesting paper on the potential for use of nanomaterials for drug delivery. I have a few questions about the manuscript.

In the literature search you make no reference to the age of the papers which are being searched. Can you mention the ages of the papers used in the study? Is there a preference for older work or more recent material? Considering the number of papers selected I am not sure that the entire time range of the study is fully covered in this case. 

I think the section on limits to the use of nanomaterials for medical applications should have more on the potential for unknown toxicological effects and the loss of material to the environment. 

page 13: why is the full reference included in here? the number would be enough.

Please review the style of the references since there is a lot of variation between how the authors are included and the use of capitals in the titles.

On a stylistic note, please remember to use subscript for all chemical formulae.

Author Response

(The authors gave the same response as above.)

Reviewer 5 Report

The review "Functionalization of Nano-particulate Drug Delivery Systems and its Influence in Cancer Therapy"  aims on the description of current trends in the design of drug delivery systems for a therapy of cancer. The text is well written, contains up to date references and in many cases also the author's critical point of view. The theme is interesting and actually important. Especially from the point of view of current trends in the treatment of cancer. 

I have only few comments:

  1. Tables 4 and 5 should be removed and its content should be described in the text. Both tables are extremely long and contain many paragraphs of text, which makes it difficult to read and understand.
  2. Chapter 6 is extremely important to understand the current state of the art of the drug delivery systems. It should be considerably extended. Authors could add future perspectives and existing and planned approaches towards solving the described limitations.
  3. Authors should also extend the chapter covering nanoparticles and extend the important answer to the question "why should the NPs be used in the first place?" 

Author Response

(The authors gave the same response as above.)

Round 2

Reviewer 1 Report

After the revision most of the issues pointed out by the reviewers were properly addressed. I just noted that the numbering in some subsections should be corrected. For example there are two subsections numbered as 4.6. For another subsection the numbering starts from 5.4.2 instead of 5.4.1.

Reviewer 3 Report

The authors responded reasonably to my comments, and the manuscript can be published.